# Assessment of the transmission blocking activity of antimalarial compounds by membrane feeding assays using natural *Plasmodium falciparum* gametocyte isolates from West-Africa

Noëlie B. Henry[1,2], Issiaka Soulama[2,3]*, Samuel S. Sermé[1,2], Judith M. Bolscher[4], Tonnie T. G. Huijs[4], Aboubacar S. Coulibaly[2], Salif Sombié[2], Nicolas Ouédraogo[2], Amidou Diarra[1,2], Soumanaba Zongo[2], Wamdaogo M. Guelbéogo[2], Issa Nébié[1], Sodiomon B. Sirima[1], Alfred B. Tiono[2], Alano Pietro[5], Katharine A. Collins[6☯], Koen J. Dechering[4☯], Teun Bousema[6☯]

1 Groupe de Recherche Action en Santé, Ouagadougou, Burkina Faso, 2 Centre National de Recherche et de Formation sur le Paludisme, Ouagadougou, Burkina Faso, 3 Institut de Recherche en Sciences de la Santé (IRSS)/CNRST, Ouagadougou, Burkina Faso, 4 TropIQ Health Sciences, Nijmegen, The Netherlands, 5 Dipartimento di Malattie Infettive, Istituto Superiore di Sanità, Roma, Italy, 6 Department of Medical Microbiology, Radboud University Nijmegen Medical Centre, Nijmegen, Netherland

☯ These authors contributed equally to this work.

* soulamacnrfp@gmail.com

**Data Availability Statement:** All relevant data are within the paper and its Supporting Information files.

## Abstract

Antimalarial drugs that can block the transmission of *Plasmodium* gametocytes to mosquito vectors would be highly beneficial for malaria elimination efforts. Identifying transmission-blocking drugs currently relies on evaluation of their activity against gametocyte-producing laboratory parasite strains and would benefit from a testing pipeline with genetically diverse field isolates. The aims of this study were to develop a pipeline to test drugs against *P. falciparum* gametocyte field isolates and to evaluate the transmission-blocking activity of a set of novel compounds. Two assays were designed so they could identify both the overall transmission-blocking activity of a number of marketed and experimental drugs by direct membrane feeding assays (DMFA), and then also discriminate between those that are active against the gametocytes (gametocyte killing or sterilizing) or those that block development in the mosquito (sporontocidal). These DMFA assays used venous blood samples from naturally infected *Plasmodium falciparum* gametocyte carriers and locally reared *Anopheles gambiae s.s.* mosquitoes. Overall transmission-blocking activity was assessed following a 24 hour incubation of compound with gametocyte infected blood (TB-DMFA). Sporontocidal activity was evaluated following addition of compound directly prior to feeding, without incubation (SPORO-DMFA); Gametocyte viability was retained during 24-hour incubation at 37°C when gametocyte infected red blood cells were reconstituted in RPMI/serum. Methylene-blue, MMV693183, DDD107498, atovaquone and P218 showed potent transmission-blocking activity in the TB-DMFA, and both atovaquone and the novel antifolate P218 were potent inhibitors of sporogonic development in the SPORO-DMA. This work establishes a

**Funding:** This work was supported by grants from the Dutch PDP fund, Medicines for Malaria Venture and Italian cooperation in Burkina Faso. Teun Bousema and Katharine A. Collins are further supported by a European Research Council (ERC) Consolidator Grant to Teun Bousema (ERC-CoG 864180; QUANTUM). The funders had no role in study design, data collection and analysis, decision to publish, or preparation of the manuscript.

**Competing interests:** No, the authors have declared that no competing interests exist.

pipeline for the integral use of field isolates to assess the transmission-blocking capacity of antimalarial drugs to block transmission that should be validated in future studies.

## Introduction

Malaria remains a significant global infectious disease, caused by parasites of the genus *Plasmodium*. In the past two decades there was a major decline in malaria cases and deaths [1]. However, this progress has recently slowed, highlighting the need for new interventions. In 2021, there were an estimated 247 million malaria cases and 619,000 malaria deaths [2].

Two important challenges for global malaria control are the emergence and spread of parasites resistant to antimalarial drugs and mosquitoes resistant to insecticides. Resistance to the cornerstone drug artemisinin has recently emerged in Africa [3]; raising serious concerns about the long-term efficacy of artemisinin-combination treatment (ACT). Efforts to reduce malaria burden and to prevent the spread of resistant parasites would benefit from strategies that specifically target malaria transmission.

The parasite responsible for malaria has a complex life cycle requiring both human and mosquito hosts. With every round of asexual parasite replication in human blood, a proportion of parasites undergo an alternative developmental pathway and transform into mosquito-transmissible male and female gametocytes. Only these male and female gametocytes, that circulate at much lower densities and peak at different times during infection than asexual parasites, are capable of infecting mosquitoes and causing onward infection. While ACTs are highly effective against the pathogenic asexual parasite stages [4] and immature gametocytes [5, 6], mature gametocytes persist after treatment [7] and can maintain malaria parasite transmission [8]. Moreover, gametocytes may be resistant to artemisinins and preferentially persist and be transmitted upon artemisinin treatment [9]. Compounds that clear or sterilize gametocytes could therefore improve the public health-impact of medication by preventing transmission shortly after treatment and reduce secondary infections.

Primaquine is one of the only compounds available that clears and sterilizes gametocytes that persist after conventional malaria treatment, and it is recommended by the World Health Organization for use in combination with ACTs in areas aiming for elimination or combating artemisinin resistance [10]. With concerns about drug resistance, it is evident that additional drugs that clear asexual parasites and also prevent transmission would be highly desirable [11, 12]. The past decade has seen a renaissance in malaria drug discovery [11, 13]. In vitro investigations on for example ATP4, PI4K or AcCS inhibitors have revealed a promising portfolio of novel antimalarials with transmission-blocking activity [12, 14, 15]. These and other transmission-blocking compounds can act by either killing or sterilizing gametocytes ('anti-gametocyte') or preventing parasite development in mosquitoes ('sporontocidal'). The evaluation of transmission-blocking drugs requires tools to reliably measure their blocking properties. Gametocyte assays are commonly used for this and are based on indicators of metabolic viability (e.g. detecting parasites with intact mitochondrial membrane potential [16], luminescence [17, 18] or metabolic activity [19]). In addition to challenges in differentiating between compound activities against male and female gametocytes [20], these assays do not directly measure transmission and may thereby fail to detect compounds with sporontocidal activity. Mosquito feeding assays are capable of comprehensively detecting transmission-blocking effects and currently primarily assess activity against *in vitro* cultured gametocytes. These cultured gametocytes are offered to receptive mosquitoes (typically *Anopheles stephensi*), either

following pre-incubation with compound prior to mosquito feeding or by adding compound directly to the gametocyte-positive bloodmeal [21]. A relevant limitation is that these mosquito feeding assays for compound screening predominantly rely on a single parasite isolate (NF54 and its clone 3D7) that was brought into culture in the 1980s [22] and is sensitive to drugs like chloroquine and sulfadoxine-pyrimethamine that have lost efficacy in the majority of malaria-endemic settings [23]. The assays thereby do not reflect the genetic and phenotypic diversity of field isolates.

Here, we developed a protocol to evaluate the transmission-blocking activity of novel compounds against *P. falciparum* field isolates. Our method allows discrimination between compounds that block transmission by acting on circulating gametocytes and compounds that interfere with parasite development in the mosquito midgut.

## Methods

### *Ex-vivo* assessments using natural gametocyte carriers from Burkina Faso

**Study area and recruitment of *P. falciparum* gametocyte carriers.** The current study comprises field activities with *ex vivo* assessments in Burkina Faso and *in vitro* assessments of compound activity against cultured gametocytes in The Netherlands (Fig 1). The field activities

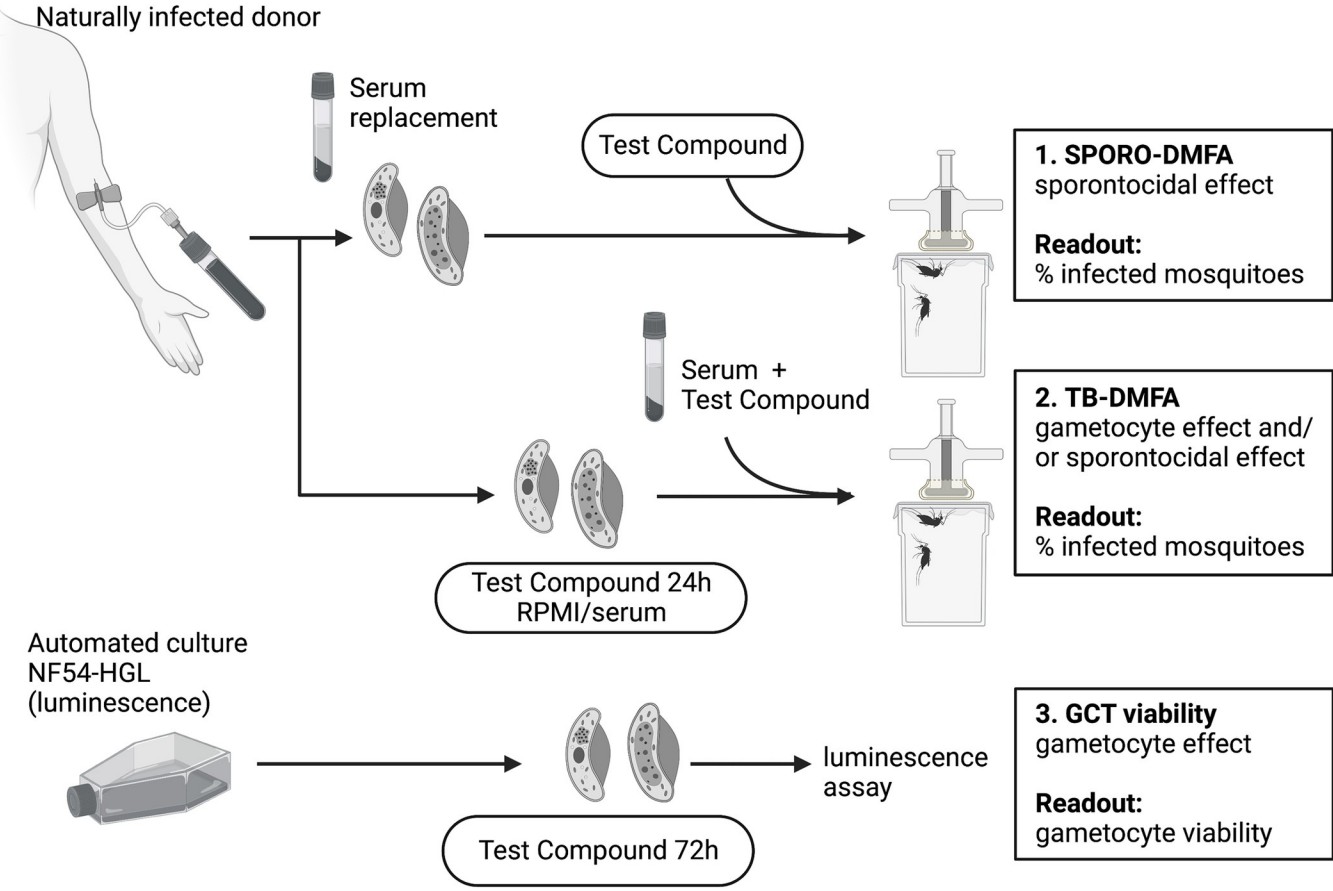

**Fig 1. Assessing the transmission-blocking effects against natural gametocyte isolates and cultured gametocytes.** Natural gametocyte isolates were used for two distinct assays detecting the overall transmission-blocking activity of compounds by incubating them with gametocyte infected blood for 24 hours (2. TB DMFA; detecting gametocyte and/or sporontocidal effects) or sporontocidal activity by directly adding compounds to a gametocyte positive blood meal just prior to feeding (1. SPORO-DMFA). Cultured gametocytes were used to test the effect of compounds on gametocyte viability following incubation (3. GCT viability). Figure created using BioRender.com.

were conducted in Saponé Health district in the province of Bazèga, located 50 km southwest of Ouagadougou, the capital city of Burkina Faso. Two surveys were conducted at schools to recruit *P. falciparum* gametocyte carriers among 5–15 year old children at the end of the malaria transmission season, i.e., from September to December 2019 and September to December 2020. Every child was clinically examined for the presence of chronic diseases, acute infections other than malaria and signs of severe malaria. Finger-prick blood was collected and used for the preparation of thick smears. Samples were considered negative if no parasites were detected in 100 microscopic fields. Both asexual and gametocyte densities were simultaneously assessed by counting against 500 leucocytes in the thick smear. Parasite counts were converted to numbers of parasites per μl by assuming a standard count of 8000 leucocytes/μL of blood. Asymptomatic malaria-infected individuals with *P. falciparum* gametocytemia ≥ 32 gametocytes/μl, were selected as blood donors for direct membrane feeding assays (DMFA), with blood collected within 24 hours of gametocyte detection. Of a total of 945 children screened, 36 met the inclusion criteria. From each of them, 9ml of whole blood was collected in lithium heparin tubes [24]; blood was stored for up to 4 hours in temperature-controlled thermos flasks with a water temperature of 35.5°C, as was previously validated to retain activity [25]. Participants received treatment according to national treatment guidelines after blood donation.

## Direct Membrane Feeding Assay (DMFA)

The DMFA was performed using female mosquitoes from an *Anopheles gambiae* colony established from field mosquitoes at Centre National de Recherche et de Formation sur le Paludisme (CNRFP) ten years ago and previously successfully used for transmission assays (e.g. [26]). Mosquitoes are maintained on 25 ± 2° C and 80 ± 10% relative humidity and fed *ad libitum* with a 5% glucose solution. For DMFA, 2–3 days old female mosquitoes were starved for ≥ 6 hours; 40 mosquitoes per cup were fed during 15–20 min via an artificial membrane (Parafilm) attached to a water-jacketed glass feeder to maintain the temperature at 37°C. After feeding, unfed mosquitoes were removed; engorged mosquitoes were kept at a temperature range from 26 to 28°C with permanent access to a glucose solution without further blood meals. Mosquito midguts were dissected 7–8 days later in 0.4% mercurochrome in phosphate buffered saline (PBS) or distilled water. The number of oocysts in the mosquito midgut was recorded to determine mosquito infection prevalence [27]. All experiments were performed in duplicate (i.e. duplicate compound exposure on two samples from the same gametocyte donor).

## Antimalarial compounds

A set of 11 compounds, being dihydroartemisin (DHA), methylene blue (MB), MMV390048 (MMV048), MMV693183, SJ773, Atovaquone, Ferroquine, Pyronaridine, DDD107498, Lumefantrine and P218, was provided by Malaria Medicine Venture for testing (MMV, Geneva, Switzerland). These compounds were dissolved in DMSO and kept in a stock solution of 10mM at -20°C. Three different concentrations equivalent to roughly 0.1x, 1x, and 10x of the mean IC50 values were tested. These IC50 values were previously determined by Standard Membrane Feeding Assay with cultured gametocytes [12]. Compounds were prepared in DMSO and then in RPMI-1640 supplemented with 25 mM sodium bicarbonate (Sigma S8761) and 10% European malaria naïve serum A (Sanquin E8813R00) to achieve a final DMSO concentration of 0.1%. Each concentration was tested in duplicate. The diluting agent for all test drugs, DMSO (Sigma-Aldrich no. D4540), was used as a negative control (referred to as "no-drug control"). Atovaquone was used as positive drug control for transmission-blockade [28].

## Optimizing the DMFA for *ex vivo* compound testing

To develop the protocols for assessment of field isolates, different incubation and DMFA conditions were evaluated: a. direct feeding of whole blood to mosquitoes on the day of collection (D0 –DIRECT), b. direct feeding of blood to mosquitoes on the day of collection following the replacement of autologous plasma with European serum A (D0 –SR), c. incubation of whole blood for 24 hours at 37°C followed by replacement of autologous plasma with European serum A just prior to mosquito feeding (D1 –Blood), d. incubation of red blood cells after replacement of autologous plasma with RPMI-1640 + 10% European serum A, followed by replacement of RPMI/serum with European serum A just prior to mosquito feeding (D1 – RPMI). Gametocyte infectivity was assessed as prevalence of mosquito infection 7–8 days after DMFA. This resulted in the following conditions:

D0 –DIRECT: 360μl of whole blood was added to 40μl 1%DMSO in RPMI1640 and mixed before mosquito feeding; the final concentration was 0.1% DMSO.

D0 –SR: Whole blood was spun 8 min at 700g at 37°C, autologous plasma removed and replaced with an equivalent volume of European malaria naïve serum A. 360μl sample was then added to 40μl 1% DMSO in RPMI1640 and mixed before mosquito feeding.

D1 –Blood: 360 μl of whole blood was added to 40 μl of 1% DMSO in RPMI1640 and incubated for 24 hours at 37°C. Hereafter, tubes were spun for 20 seconds at 14000 rpm and RPMI1640 was carefully removed and replaced by an equivalent volume of 0.2% DMSO in European malaria naïve serum A. Tubes were mixed before mosquito feeding.

D1 –RPMI: Donor plasma was replaced by RPMI1640 supplemented with 25 mM sodium bicarbonate (Sigma S8761) and 10% European malaria naïve serum A. An aliquot of 360 μl of sample was added to 40 μl of 1% DMSO in RPMI1640 and incubated for 24 hours at 37°C. Hereafter, tubes were spun for 20 seconds at 14000 rpm and RPMI1640 was carefully removed and replaced by an equivalent volume of 0.2% DMSO in European malaria naïve serum A. Tubes were mixed before mosquito feeding.

## Preparation of blood for the anti-sporogony DMFA (SPORO-DMFA)

To test the effect of compounds when added directly to the blood sample prior to feeding–and thus their direct effect on sporogony—each sample was spun 8 min at 700g and 37°C to remove autologous plasma and replace it with an equivalent volume of European malaria naïve serum A. An aliquot of 360μl of each sample was added to 40μl of 10x concentrated compound solution in RPMI1640 and mixed before mosquito feeding.

## Preparation of blood for the transmission-blocking DMFA (TB-DMFA)

To allow for 24 hour incubation with compounds of interest, donor plasma was replaced by RPMI1640 supplemented with 25 mM sodium bicarbonate (Sigma S8761) and 10% European malaria naïve serum A. An aliquot of 360 μl of each sample was added to 40 μl of 10x concentrated compound solution in RPMI1640 and incubated for 24 hours at 37°C. Hereafter, tubes were spun for 20 seconds at 14000 rpm and RPMI1640 was carefully removed and replaced by an equivalent volume of 2x concentrated compound in European malaria naïve serum A. Tubes were mixed before mosquito feeding.

### *In vitro* gametocyte viability experiments using cultured *Plasmodium* parasites

Experiments with cultured gametocytes were conducted at TropIQ in Nijmegen. For this, a previously described high throughput luminescence assay was used to monitor the viability of gametocytes [29]. *P. falciparum* NF54-HGL parasites were cultured in RPMI 1640 medium

supplemented with 367 mM hypoxanthine, 25 mM HEPES, 25 mM sodium bicarbonate and 10% human type A serum in an automated large-scale culture system. Cultures were setup at 1% parasitemia and treated with 50 mM N-acetyl-D-glucosamine from day 4 to day 8 to kill asexual parasites. On day 11 stage III-IV gametocytes were isolated by discontinuous 63% Percoll gradient centrifugation. The purity of the resulting gametocyte fraction was determined by microscopy and revealed the presence of 77% residual red blood cells. Determination of the gametocyte differentiation stage was performed by microscopic examination of Giemsa stained thin smears following the classification proposed by Hawking et al. [30] 5,000 gametocytes were seeded per well in 30 μl in white 384-well plates containing 30 μl of compounds diluted in medium (in a final concentration of 0.1% DMSO). After 72h incubation, 30 μl of ONE-Glo reagent (Promega) was added and luminescence was quantified using the BioTek Synergy 2 Plate reader. Values were normalized to DMSO- and DHA-treated controls, as previously described [29].

## Data analysis

For gametocyte viability, data were expressed as the percentage effect relative to the MIN (1 μM dihydroartemisinin) and MAX (0.1% DMSO) controls. For DMFAs, data were expressed relative to the negative (vehicle) controls for oocyst prevalence (i.e., the proportion of infected mosquitoes). Mosquito infection prevalence and infection intensity (oocyst density) are strongly correlated. Since infection prevalence is not saturated in experiments with natural gametocyte carriers, there is limited information in oocyst density and all analyses were based on oocyst prevalence (i.e. the proportion of infected mosquitoes. Pairwise comparisons on the proportion infected mosquitoes were performed by Wilcoxon matched pairs signed-rank test. IC50 values were calculated by applying a four-parameter logistic regression model using a least-squares method to find the best fit using the Graph pad Prism 8.3.0 software package.

## Ethics statement

The protocols for human blood collection and for mosquito maintenance were approved by the CNRFP Institutional Ethics Committee (N˚2019-06-000004) and the national Ethics Committee (N˚2020-01-006). Before enrollment, written informed consent was obtained from each volunteer and/or their legal guardian.

## Results

### Development and validation of assays to allow evaluation of transmission-blocking compounds against natural field isolates

Gametocytes were collected from naturally infected individuals and transported to the lab in a temperature-controlled thermos flask [25]. To evaluate the viability of gametocytes after overnight incubation, mosquito infection rates in DMFAs performed after incubation were compared to mosquito infection rates when whole blood was fed directly to mosquitoes on the same day as collection (D0 –Direct). Incubating gametocyte-infected blood during 24 hours (D1 –Blood) resulted in a marked loss of gametocyte viability for all three samples that were infectious to mosquitoes on the day of sampling (Fig 2A). Removing autologous plasma and incubating gametocytes in RPMI-1640/10% European malaria naïve human serum A for 24 hours (D1 –RPMI) retained gametocyte infectivity following incubation without an apparent reduction in mosquito infection rates (Fig 2B). This incubation condition with RPMI/serum was considered to be the optimal condition for further experiments with 24-hour incubation with compounds. Since with these conditions autologous plasma was replaced with European serum A, a similar replacement procedure was used for DMFA experiments without

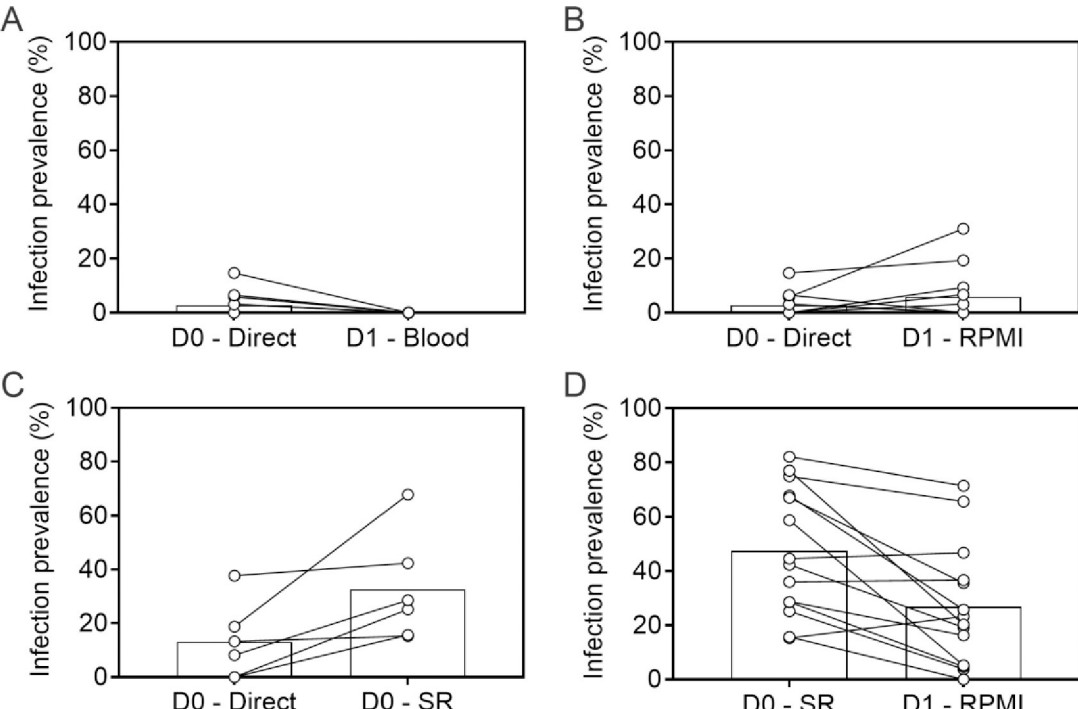

**Fig 2. Optimization of the DMFA protocol and incubation conditions.** Mosquito feeding assays were either performed with either gametocyte infected whole blood on day of collection (D0 –Direct), with gametocyte infected whole blood following 24 hours incubation (D1 –Blood), with gametocyte infected blood after the donor plasma was removed and replaced with malaria naïve serum on the day of blood collection (D0 –SR), or with gametocyte infected red blood cells that had been incubated for 24 hours in RPMI that was replaced with malaria naïve serum before feeding mosquitoes (D1 –RPMI). The graphs present the comparison of the proportion of infected mosquitoes for A) the comparison of immediate feeding of whole blood at D0 (D0-Direct) compared to D1 (D1-Blood); B) the comparison of immediate feeding of whole blood at D0 (D0-Direct) compared to 24-hour incubation with RPMI/serum (D1-RPMI); C) the comparison of immediate feeding of whole blood at D0 (D0-Direct) compared to feeding on the same day with serum replacement (D0-SR); D) the comparison of feeding of whole blood following serum replacement at D0 (D0-SR) compared feeding after 24 hours of incubation with RPMI/serum (D1-RPMI). Symbols indicate individual donors; lines connect experiments performed on the same blood aliquot. Bars indicate the mean infection prevalence.

incubation, i.e. experiments where gametocytes were directly offered to mosquitoes. Replacing autologous plasma with European serum A (D0 –SR) resulted in a small increase in gametocyte infectivity (Fig 2C). In conclusion, the final conditions were as follows: for compound screening in the DMFA that was performed on the day of phlebotomy (i.e. SPORO-DMFA, where compound would be directly added to gametocytes without incubation), autologous plasma was replaced with European malaria naïve serum A; for compound screening with 24-hour gametocyte incubation, RPMI/European serum A was used for 24 hours with replacement of the RPMI with European serum A before mosquito feeding (TB-DMFA). Comparison of this last condition (D1 –RPMI) with an immediate feed (D0 –SR) showed a small reduction in gametocyte infectivity following the 24hr incubation, but infectivity levels were sufficiently high to enable compound evaluation (Fig 2D).

## Sporontocidal effects of antimalarial compounds against *Plasmodium falciparum* field isolates

Following the above-described assay optimization, we evaluated the activity of 9 compounds (Atovaquone, DHA, Methylene blue, MMV390048, DDD107498, P218, Pyronaridine,

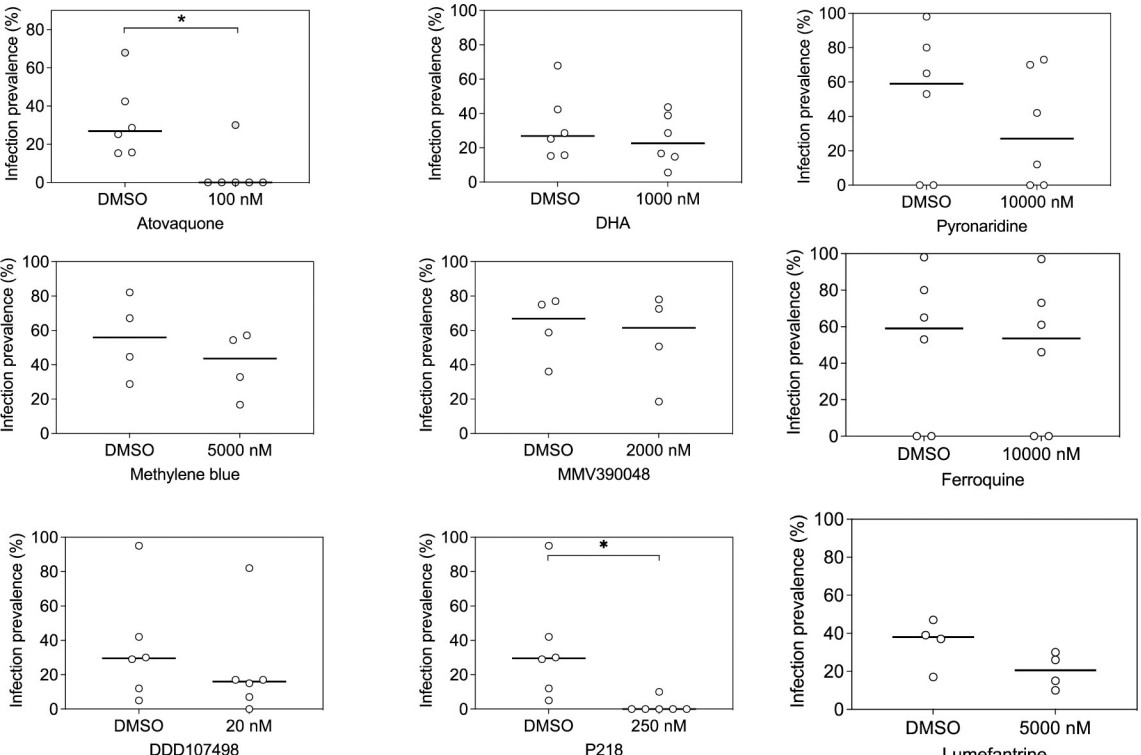

**Fig 3. Transmission-blocking effects of compounds when directly added to *Plasmodium falciparum* gametocyte field isolates prior to feeding.** Effects of Atovaquone, DHA, Pyronaridine, Methylene blue, MMV390048, Ferroquine, DDD107498, P218 and Lumefantrine on the proportion of mosquitoes that became infected after feeding on gametocytes from naturally infected gametocyte donors where serum was replaced and the compound was added to the blood meal immediately prior to feeding. Every symbol represents an individual gametocyte donor whose blood was offered to mosquitoes with DMSO control (0.1% DMSO) or the compound at a single concentration. Lines connect experiments performed on the same blood aliquot; bars indicate the mean infection prevalence. The asterisks indicate statistical significance (p<0.05) in a Wilcoxon matched-pairs signed-rank test. Of note, the number of observations is small so lack of statistical significance should not be interpreted as evidence of no difference. Experiments with no infected mosquitoes in the DMSO control were not included in the statistical analyses.

Ferroquine, and Lumefantrine) that were added directly (without incubation) to the gametocyte positive blood meal with serum replacement. These experiments were performed with blood from 4–6 gametocyte positive donors per compound. The vehicle control for compounds (0.1% DMSO in RPMI) was used as a negative control. These experiments showed that Atovaquone almost completely inhibited infectivity at 100nM and also demonstrated high potency of P218. With our limited number of paired feeds, we observed no evidence for a statistically significant effect of DHA, Methylene blue, MMV390048, DDD107498, Pyronaridine, Ferroquine, or Lumefantrine on gametocyte infectivity at micromolar concentrations (Fig 3). This confirms the potent sporontocidal effect of Atovaquone [31] while also demonstrating that an established transmission-blocking drug like Methylene blue exerts its effect through an anti-gametocyte mechanism [32].

## Transmission-blocking activity of antimalarial compounds against *Plasmodium falciparum* field isolates

The transmission-blocking activity of 11 marketed and experimental antimalarial compounds was evaluated by incubating compounds with gametocytes in RPMI/European malaria naïve serum A for 24 hours before being fed to mosquitoes. This experimental set-up detects the

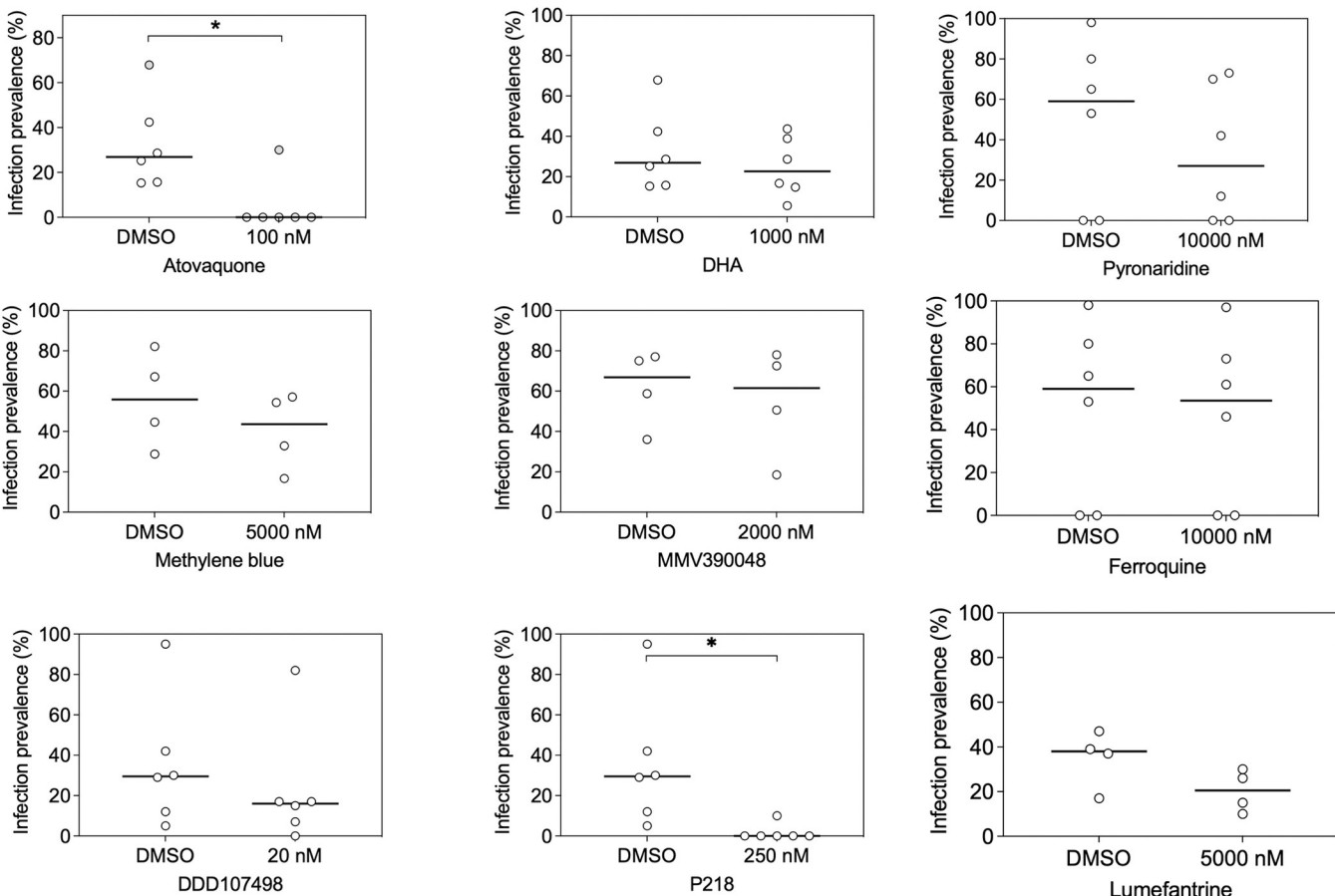

**Fig 4. Transmission-blocking effects of compounds when incubated with *Plasmodium falciparum* gametocyte field isolates.** Serial dilutions of compounds were added to gametocytes in RPMI/European serum A and incubated for 24 hours prior to feeding. The plot indicates the effects on mosquito infection prevalence of DHA, Methylene Blue, SJ733, MMV390048, MMV693183, DDD107498, P218, Pyronaridine, Ferroquine, Lumefantrine, and Atovaquone. DMSO was used as a negative control. Symbols indicate individual donors with connecting lines indicating the different incubation conditions for the same donor. Error bars indicate standard deviations from technical replicates (n = 2); some compounds were only tested in with a single technical replicate and therefore no error bars are provided.

effect of compounds against gametocytes and/or its sporontocidal effects. All compounds were tested at 3 different dilutions that reflected best estimates of 0.1xIC50, 1xIC50 and 10xIC50 based on prior assessments using *in vitro* cultured gametocytes [21]. The baseline infection rate was determined by control DMFAs(incubation with 0.1% DMSO) in every experimental run. For each test performed, 4 to 6 gametocyte carriers were used. All controls showed mosquito infection rates ranging from 5 and 75%, in line with data presented in Fig 2, confirming that field-derived gametocytes retained infectivity after 24 hours incubation in RPMI1640 supplemented with 10% human malaria naïve serum. All compounds tested, with the exception of DHA and Ferroquine, reduced mosquito infection rates in a dose dependent manner and this was observed for all gametocyte isolates tested (Fig 4). From these data, we estimated the IC50s of tested compounds by averaging the data from different isolates (Table 1). In this same table, IC50 values from prior assessments with cultured *P. falciparum* NF54 gametocytes are presented. Most of our field-based IC50 estimates were within one log from values obtained from cultured NF54 gametocytes. Exceptions were DHA, that was not active against the tested field isolates, and compound P218 that appeared more potent against field isolates, as compared with NF54.

**Table 1. Comparison of activity of compounds tested by incubating naturally acquired and cultured gametocytes.** The inhibitory concentration (IC50) is presented for compounds that were incubated for 24 hours with gametocytes from naturally infected gametocyte carriers (field isolates) or cultured NF54 gametocytes (NF54). IC50 values are based on Fig 3 for field isolates or on published data with cultured NF54 gametocytes from the same study team. Alongside these IC50 values in mosquito feeding assays, IC50 values from NF54 gametocyte viability assays are presented. ND = not done; * = data taken from [21]; ** = data taken from [33]. Experiments with no infected mosquitoes in the DMSO control were not included in the dose response analyses.

| | IC50 (nM) | | |
|---|---|---|---|
| Compound | SMFA | TB-DMFA | Gametocyte viability assay |
| | NF54 | Field isolates | NF54 |
| Dihydroartemisinin (DHA) | 93 | >1000 | 40 |
| Methylene Blue | 100 | 68 | 35 |
| MMV048 | 174 | 218 | 41 |
| MMV693183 | 38 | 167 | 34 |
| SJ733 | 1023 | 865 | 830 |
| Atovaquone | 2 | <0.2 | >5,000* |
| Ferroquine | 850 | >10,000 | ND |
| Pyronaridine | 1000 | 8,640 | >1,000* |
| DDD107498 | 2 | 16 | 2** |
| Lumefantrine | 427 | 475 | >1,000* |
| P218 | 25 | <2.5 | >5,000* |

## Anti-gametocyte activity of antimalarial compounds against *Plasmodium falciparum* culture strain NF54

To compare the activity of the compounds against NF54 lab parasites and field isolates to provide a quantitative comparison of dose-dependent effects, where data were not already available we performed a full dose response gametocyte viability assay (with the exception of Ferroquine). All compounds, with the exception of atovaquone, fully reduced gametocyte viability in a dose-dependent manner (Fig 5). Comparison of IC50 values between gametocyte viability assay, TB-DMFA using field isolates and SMFA using NF54 laboratory strain shows broad agreement between the assays for Methylene Blue, MMV048, MMV693183, SJ733, Pyronaridine, DDD107498 and Lumefantrine while other compounds (notably DHA) showed marked differences in activity against NF54 and field isolates.

## Discussion

The identification of compounds that block transmission of *P. falciparum* requires assessments that offer both throughput and definitive evidence on functional transmission-blocking effects. Acknowledging the limitations of *in vitro* screening against cultured gametocytes that have limited genetic diversity, we here present a methodology where we expose naturally acquired *P. falciparum* gametocytes from Burkina Faso to test compounds *ex vivo* and assess their infectivity to mosquitoes.

Assays to determine *ex vivo* susceptibility to antimalarial drugs have become a standard for asexual parasites, allowing the detection of variation or temporal changes in drug sensitivity [34]. Despite the importance of transmission-blocking properties of antimalarials, similar assays have been unavailable for gametocytes and require a different approach with access to an insectary with established mosquito feeding assays. In the current study, we aimed to establish assays that allow testing of transmission-blocking compounds against genetically diverse gametocytes, as are acquired by naturally exposed individuals. In optimizing our assay, we observed that storing gametocyte-infected blood for 24 hours at 37°C resulted in a loss of gametocyte infectivity; this infectivity was largely restored by replacement of plasma with RPMI1640/10% European serum A during incubation. While also with RPMI/10% European

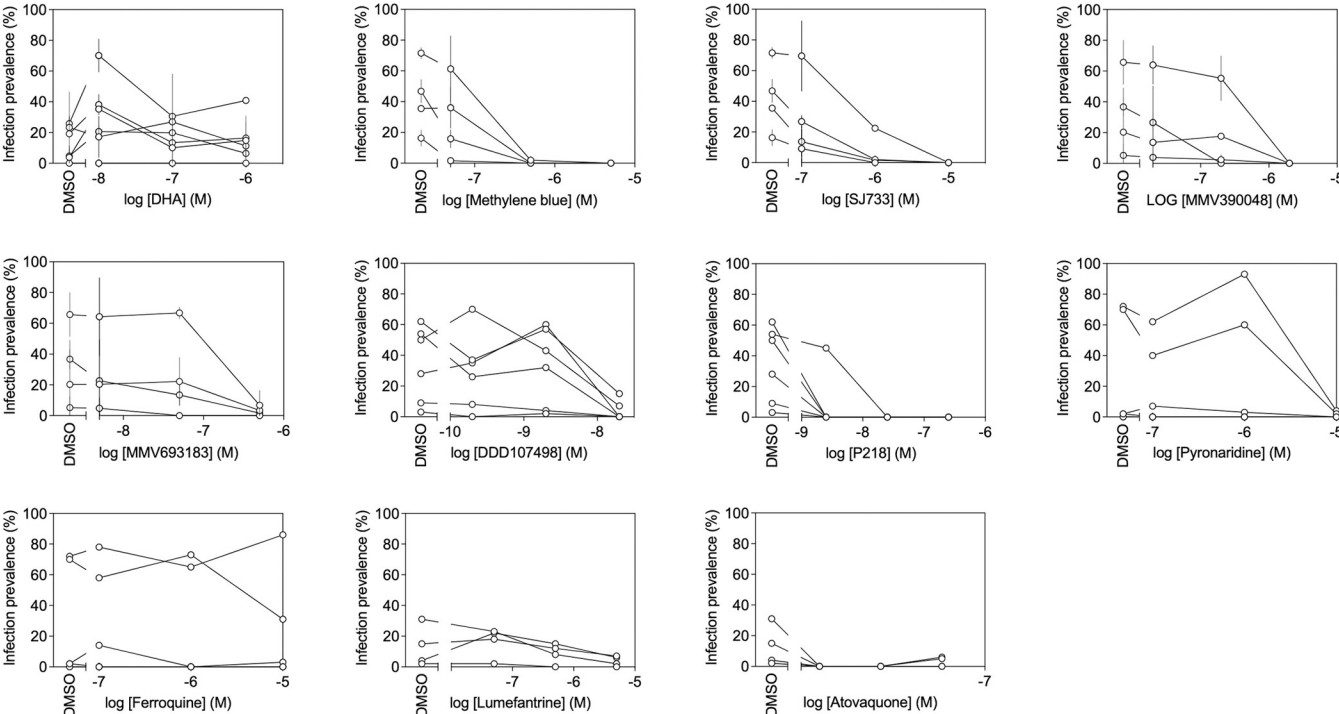

**Fig 5. Effect of transmission-blocking compounds when incubated with cultured *Plasmodium falciparum* gametocytes.** Serial dilutions of compounds were added to cultured gametocytes and incubated for 72 hours upon which gametocyte viability, determined by luminescence, was calculated at percentage of DMSO control. The graphs indicate the effect of DHA, Methylene blue, SJ733, MMV390048, MMV693183 and Atovaquone on gametocyte viability. Error bars indicate standard deviation. Error bars indicate standard deviations.

serum incubation conditions a modest reduction in mosquito infection rates was observed, the majority of gametocyte donors remained infective to mosquitoes at levels that allowed examination of possible transmission-blocking effects of compounds. We were thus able to screen compounds for their ability to reduce mosquito infection rates in a direct manner (i.e. sporontocidal activity that directly affects infectivity when compounds are co-ingested with gametocytes) or in a more indirect manner (killing or sterilizing gametocytes prior to mosquito uptake). These two different mechanisms by which compounds can reduce transmission are relevant from a public health perspective. Compounds that render gametocytes non-infectious, i.e. block the gametogenesis capacity of gametocytes, can have a non-reversible sterilizing effect on infections and persists after drug levels have waned. This is distinct from a direct effect where a compound is active when present in the mosquito blood meal. Prior assessments with *in vitro* cultured gametocytes have indeed demonstrated that drugs may differ in their anti-gametocyte and sporontocidal activity [35]. For example, atovaquone potently blocks oocyst formation in the mosquito but has no effect on gametocyte viability or gamete formation [21].

In our assays where compounds were directly added to a gametocytemic blood meal (our SPORO-DMFA), we observed that atovaquone has pronounced sporontocidal activity [36]. This is in line with previous observations that atovaquone inhibits ookinete formation [37] and affects transmission even if mosquitoes imbibe the compound several days after having taken up infectious gametocytes [31]. Atovaquone completely prevented transmission of all but one isolate when tested at 100 nM. Unfortunately, we were not able to characterize this isolate. Without pre-incubation of a compound with gametocytes, DHA, MMV390048, DDD107498, Pyronaridine, Ferroquine, Lumefantrine and methylene blue had very limited effects on the infectivity of field gametocytes. Although the number of gametocyte donors was

modest and subtle effects cannot be ruled out, our data indicate that these compounds do not have a marked direct effect on gametogenesis, fertilization and early parasite development in the mosquito midgut. This is in line with previous data obtained with laboratory strains [14, 21]. When compounds were incubated with gametocytes for 24 hours, we observed marked transmission-blocking effects of Methylene blue, MMV390048, MMV693183 and DDD107498 at concentrations below 500nM. Methylene blue is one of the oldest synthetic antimalarial drug registered for clinical use, with a potent anti-gametocyte effect *in vivo* and *in vitro* on all gametocyte stages [5, 38]. It inhibited infectivity of field isolates with an IC50 of 68 nM that is comparable to results obtained against the NF54 laboratory strain [39]. MMV693183 is a pantothenamide acetylCoA synthetase inhibitor from the MMV portfolio that has *in vitro* transmission-blocking activities [29]. This is confirmed by the data presented here although the potency against field isolates appears slightly lower than observed against NF54 laboratory parasites. MMV390048 reduced viability of gametocytes in culture and also inhibited oocyst formation from cultured gametocytes at a concentration of 111nM [14]. In our study MMV390048 showed complete inhibition of parasite transmission with an estimated IC50 of 218 nM. We also confirmed the high transmission-blocking potency of the novel anti-folate P218 [40] and DDD107498 [33] that targets the translation elongation factor 2 (eEF2) that is essential for protein synthesis and observed transmission blocking activity.

DHA was not active in reducing transmission, neither when added directly to the blood meal or when gametocytes were incubated with this compound. When laboratory-cultured NF54 gametocytes are incubated with DHA, transmission is markedly reduced even though this requires higher concentrations than required for asexual stage parasites [41]. The lack of activity of DHA in the current study contrasts not only with findings with cultured gameto-cytes but also with findings from a recent study from Mali that determined transmission-blocking activity of compounds against gametocyte field isolates. In that study, undertaken around the same time as the current work, colleagues developed a methodology that is similar to the one presented here: gametocyte isolates were exposed to compounds for 48 hours and, after replacement of compound-containing medium with horse serum, offered to mosquitoes [42]. This study observed marked reductions in mosquito infection rates when gametocytes were incubated with DHA at 1μM, and also confirmed the transmission-blocking properties of primaquine and novel compounds including PI(4)K-inhibitor KDU691 and imidazolopipera-zine GNF179 [42]. Of note, in Mali both the medium conditions (10% horse serum at 4% hematocrit) and duration of compound exposure (48 hours) where slightly different from ours. Likewise assays evaluating activity against NF54 use a longer duration of incubation (i.e. 72 hours). Future studies should determine whether there is indeed lower efficacy of DHA against gametocytes from Burkina Faso, or if differences in incubation conditions or exposure duration are responsible for the apparent discrepancy.

Both the current study and the independent study from Mali demonstrated that viability and infectivity of gametocytes can be retained *ex vivo* with optimised culture conditions that include a source of malaria-naïve serum [42]. The increased infectivity that we observed after replacing autologous plasma with malaria naïve serum is commonly observed and may be related to the removal of transmission-blocking malaria antibodies or non-specific effects of blood factors [42–44]. Our findings further indicate that while incubation for 24 hours results in a small loss of gametocyte infectivity, mosquito infection rates are still sufficiently high to enable evaluation of compounds and differentiate between compounds with high- and with low transmission-blocking activity. Our current study tested a limited set of concentrations per compound. Although this allowed a broad comparison between transmission-blocking effects observed against field isolates versus laboratory studies, a more detailed analyses of more subtle changes in potency would benefit from more extensive dose-response analyses.

## Conclusion

In conclusion, this study demonstrated the establishment of a protocol for the use of field *P. falciparum* gametocyte isolates to test novel antimalarial compounds. we observed transmission-blocking effects on field isolates were broadly in line with results from laboratory strain NF54. A notable exception was dihydroartemisinin that was not active against field isolates. This highlights the importance of including field isolates when evaluating the transmission-blocking properties of novel antimalarial drugs.

## Supporting information

**S1 Data. Data underlying the figures are provided as supplemental data.**
(XLSX)

## Acknowledgments

We would like to thank all the institutional staff for their contribution to and support for the study. We are grateful to the children, their parents or guardians for their participation to this study. We also thank the ISS, TropIQ and Radboudumc teams for technical support.

## Author Contributions

**Conceptualization:** Noëlie B. Henry, Issiaka Soulama, Judith M. Bolscher, Tonnie T. G. Huijs, Aboubacar S. Coulibaly, Issa Nébié, Alano Pietro, Katharine A. Collins, Koen J. Dechering, Teun Bousema.

**Data curation:** Noëlie B. Henry, Issiaka Soulama, Judith M. Bolscher, Katharine A. Collins, Koen J. Dechering.

**Formal analysis:** Noëlie B. Henry, Issiaka Soulama, Judith M. Bolscher, Alano Pietro, Katharine A. Collins, Koen J. Dechering.

**Funding acquisition:** Issiaka Soulama, Issa Nébié, Sodiomon B. Sirima, Alfred B. Tiono, Alano Pietro, Katharine A. Collins, Koen J. Dechering, Teun Bousema.

**Investigation:** Noëlie B. Henry, Issiaka Soulama, Samuel S. Sermé, Judith M. Bolscher, Nicolas Ouédraogo, Wamdaogo M. Guelbéogo, Issa Nébié, Koen J. Dechering.

**Methodology:** Noëlie B. Henry, Issiaka Soulama, Samuel S. Sermé, Judith M. Bolscher, Tonnie T. G. Huijs, Aboubacar S. Coulibaly, Amidou Diarra, Wamdaogo M. Guelbéogo, Issa Nébié, Alano Pietro, Katharine A. Collins, Koen J. Dechering.

**Project administration:** Issiaka Soulama, Wamdaogo M. Guelbéogo, Sodiomon B. Sirima, Katharine A. Collins, Koen J. Dechering.

**Resources:** Noëlie B. Henry, Issiaka Soulama, Samuel S. Sermé, Judith M. Bolscher, Tonnie T. G. Huijs, Aboubacar S. Coulibaly, Salif Sombié, Nicolas Ouédraogo, Amidou Diarra, Soumanaba Zongo, Wamdaogo M. Guelbéogo, Issa Nébié, Alfred B. Tiono, Katharine A. Collins, Koen J. Dechering.

**Software:** Katharine A. Collins, Koen J. Dechering.

**Supervision:** Noëlie B. Henry, Issiaka Soulama, Wamdaogo M. Guelbéogo, Katharine A. Collins, Koen J. Dechering.

**Validation:** Noëlie B. Henry, Issiaka Soulama, Samuel S. Sermé, Judith M. Bolscher, Tonnie T. G. Huijs, Katharine A. Collins, Koen J. Dechering.

**Visualization:** Noëlie B. Henry, Issiaka Soulama, Samuel S. Sermé, Judith M. Bolscher, Aboubacar S. Coulibaly, Wamdaogo M. Guelbéogo, Issa Nébié, Sodiomon B. Sirima, Alfred B. Tiono, Katharine A. Collins, Koen J. Dechering, Teun Bousema.

**Writing – original draft:** Noëlie B. Henry, Issiaka Soulama, Samuel S. Sermé, Tonnie T. G. Huijs, Aboubacar S. Coulibaly, Salif Sombié, Nicolas Ouédraogo, Amidou Diarra, Soumanaba Zongo, Wamdaogo M. Guelbéogo, Issa Nébié, Sodiomon B. Sirima, Alfred B. Tiono, Alano Pietro, Katharine A. Collins, Koen J. Dechering, Teun Bousema.

**Writing – review & editing:** Noëlie B. Henry, Issiaka Soulama, Samuel S. Sermé, Judith M. Bolscher, Tonnie T. G. Huijs, Aboubacar S. Coulibaly, Salif Sombié, Nicolas Ouédraogo, Amidou Diarra, Soumanaba Zongo, Wamdaogo M. Guelbéogo, Issa Nébié, Sodiomon B. Sirima, Alfred B. Tiono, Alano Pietro, Katharine A. Collins, Koen J. Dechering, Teun Bousema.

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
