## [Decision Letter · Decision Letter 0]

7 Sep 2022

PONE-D-22-17171Assessment of the transmission blocking activity of antimalarial compounds by membrane feeding assays using West-African patient-derived Plasmodium falciparum gametocytesPLOS ONE

Dear Dr. SOULAMA,

Thank you for submitting your manuscript to PLOS ONE. After careful consideration, we feel that it has merit but does not fully meet PLOS ONE’s publication criteria as it currently stands. Therefore, we invite you to submit a revised version of the manuscript that addresses the points raised during the review process. Two experts who know membrane feeding experiments very well commented that this work is very important but the information in this manuscript is not enough for the readers. So, I also encourage the Authors to consider all the Reviewers' comment for the improvement. Please also provide point-by-point response. Please submit your revised manuscript by Oct 13 2022 11:59PM. If you will need more time than this to complete your revisions, please reply to this message or contact the journal office at plosone@plos.org. Please include the following items when submitting your revised manuscript:A rebuttal letter that responds to each point raised by the academic editor and reviewer(s). You should upload this letter as a separate file labeled 'Response to Reviewers'.A marked-up copy of your manuscript that highlights changes made to the original version. You should upload this as a separate file labeled 'Revised Manuscript with Track Changes'.An unmarked version of your revised paper without tracked changes. You should upload this as a separate file labeled 'Manuscript'.

We look forward to receiving your revised manuscript.

Kind regards,

Takafumi Tsuboi

Academic Editor

PLOS ONE

Journal Requirements:

No, the funders had no role in study design, data collection and analysis, decision to publish, or preparation of the manuscript.

This work was supported by grants from the Dutch PDP fund, Medicines for Malaria Venture and Italian cooperation in Burkina Faso. The funders had no role in study design, data collection and analysis, decision to publish, or preparation of the manuscript.

No, the funders had no role in study design, data collection and analysis, decision to publish, or preparation of the manuscript.

4. Please include your tables as part of your main manuscript and remove the individual files. Please note that supplementary tables (should remain/ be uploaded) as separate "supporting information" files

Reviewers' comments:

Reviewer's Responses to Questions

**Comments to the Author**

1. Is the manuscript technically sound, and do the data support the conclusions?

Reviewer #1: Partly

Reviewer #2: Yes

2. Has the statistical analysis been performed appropriately and rigorously? 

Reviewer #1: No

Reviewer #2: N/A

3. Have the authors made all data underlying the findings in their manuscript fully available?

Reviewer #1: No

Reviewer #2: Yes

4. Is the manuscript presented in an intelligible fashion and written in standard English?

Reviewer #1: Yes

Reviewer #2: No

5. Review Comments to the Author

Reviewer #1: The authors developed an indirect membrane feeding assay (indirect-MFA), where researchers can test transmission-blocking activity of antimalarial drugs with field gametocytes after 24-hour incubation, and compared IC50 of several drugs between the indirect-MFA and standard membrane feeding assay (SMFA) with laboratory-adapted NF54 parasites (a gold standard method at this moment). While their scientific approach is reasonable, the major results are missing in the current manuscript; i.e., the authors did not submit Table 1. Therefore, this reviewer cannot assess whether their statements are supported by the data. Furthermore, while the development of indirect-MFA is one of major points of this manuscript, the method section is not satisfactory. Thus, it is difficult for other researchers to use the new assay.

Major comments;

1) Please submit Table 1, and show statistical results (e.g., Pearson correlation coefficients and p-values) to support the conclusion; good agreement in IC50 among Gametocyte viability assay, indirect-MFA and indirect-SMFA (in Line 266-268).

2) Method for indirect-MFA

Please describe the method in detail so that readers can use the assay. More specifically;

During the 24-h incubation

(a) medium; Was it pure RPMI (based on Line 202), or same as NF54 culture medium (including hypoxanthine, HEPES and sodium bicarbonate; Line 158-160), or something else?

(b) container; flask, tube or plate to keep the infected blood

(c) hematocrit; was the blood diluted at a certain ratio, or same ratio of RBC/serum in the original blood?

(d) IC50 was pre-determined to set 0.1x, 1x and 10x IC50 concentrations. But it is not clear how IC50 for each drug was estimated; IC50 in prevalence of infected mosquitoes or that for oocyst density? What assay (and which strain of parasites) was used to determine the value? Did you use a value from publication (if so, cite the paper), or did you use your own data?

For the feed,

(e) composition of feeding samples; If 100% of the liquid part was replaced by a malaria naïve serum after the 24-h incubation, then there was no room to mix the test drug. Was it a mixture of XX%v/v of human serum and XX%v/v of drug solution in RPMI? And what was the hematocrit of final feeding samples?

Minor comments;

3) Can Indirect-MFA check gametocytocidal activity?

Since there was no evaluation for gametocytemia or exflagellation after the 24-h drug treatment, from the lower infection prevalence, we cannot tell whether the drug has sporontocidal activity, gametocytocidal activity, sterilizing gametocytes, or mixture of them. For example, if a drag kills gametocytes during the 24-h incubation (gametocytocidal), the final gametocytemia should be lower than that in the starting blood, but it was not evaluated. On the other hand, a drug may just reduce female fertilization activity (which is technically very difficult to measure) without killing them. In any case, it is impossible to tease out the mechanism of action from the described indirect-MFA method. Please fix all related text in Introduction, Results and Discussion sections.

4) “overnight” incubation?

People usually think “overnight” means ~10-12-h, not 24-h. To avoid the confusion, please do not use a word of “overnight” throughout the paper.

5) Gametocyte viability assays

Based on the method section (Line 161-163), two types of gametocytes (early and late gametocytes) were prepared. But I guess Gametocyte viability assays were performed only using late gametocytes. If so, where did the authors use the “early gametocyte”?

In addition, for clarity, please specify that the assay was done with NF54, such as, we performed full dose response gametocyte viability assays with “NF54” for selected compounds (Line 263-264).

6) What is “indirect SMFA”?

In Line 254, a term of “SMFA” is written, but another term “indirect SMFA” is seen in Line 267. Are they the same? If so, please use the same terminology to avoid a confusion. If different, please use different words, e.g., “direct SMFA” and “indirect SMFA”. In addition, I guess “indirect SMFA” means that an assay where NF54 parasites are pre-incubated with a test drug for 24 h before feed, but please clarify.

7) Bars and error bars

For all figures, please explain what bars (e.g., average, median, geometric mean) and error bars (e.g., sd, sem, 95%CI) mean.

8) Fig 4

Same as Fig 3, “paired” data points (same donor’s blood were treated with DMSO or drug for 24-h) should be linked. Without the lines, readers cannot interpret the results. For example, the authors conclude that no inhibition by Atovaquone for one isolate (Line 312-313). But if the infection rate for the one isolate changed from ~70% (the highest point in DMSO) to ~30% (the highest point in 100 nM drug), then the conclusion should be different. In addition, a paired test should be used for the statistical analysis (please specify the name of statistical test). Once the authors do so, I’m afraid conclusion may change, e.g., there could be a significant reduction by Methylene blue (and also by DHA). Please revise the related text if needed.

9) Typo

Line 173; 10 to the power of 4, not 104

Line 173, 174 and 176; 30 microliter, not 30 mL

Line 247; to determine the “infection rates”, not “oocyst intensities” (no TRA data in this paper).

Line 307, take out “on”

Line 320 and 334; the format of citation is wrong.

Reviewer #2: This is paper describes a method to determine transmission reducing activity of test compounds against P. falciparum isolates. Overall the authors have done an admirable job. Testing field isolates are logistically challenging but very important.

I have comments below to improve the manuscript and make it more transparent to readers.

1) The writing could be polished by a native English user. In several places, the tense is in appropriate and the article is missing. Words such as mosquitoes/mosquitos or transmission blocking/transmission-blocking should be harmonized throughout the manuscript.

2) There is no description of how the transmission blocking experiment was done on 3D7/NF54 gametocytes (i.e. the Indirect-SMFA). I am not sure if the data are original to the study, or referred to published work. Please provide the details of this experiment in the method (if original data), or in the discussion (if from previous studies).

I ask this because it is not clear whether indirect-SMFA is comparable to Indirect-MFA. Did the experiment use enriched gametocytes, and if so, early gametocytes or late gametocytes? Was the treatment also for 24 hours in the same culture medium? Without these details, it’s difficult to know whether the discrepancy in the results of 3D7/NF54 vs field isolates was due to biological (i.e. isolate-to-isolate) differences or the technical aspects of the assays.

3) Line 109: Were the participants treated for malaria after providing 9 ml blood?

4) Figures 1 & 2 are not clear. The arrow after the 24 hour incubation should point to the mosquito feeding cup.

5) Malaria naïve serum used for a) the 24 hr culture medium or b) plasma/RPMI replacement before MFA: was it from an AB blood group donor? Please clarify this in the method section.

6) Line 116: “In D0 DMFA” Should this be D1 DMFA?

7) Under the method section: Please add statement that, for Indirect DMFA, naïve serum containing the test compound at the test concentration was used to replace the medium before membrane feeding.

8) Keywords: Anopheles coluzzii? I thought the experiments used An. gambiae throughout.

9) Although the term ‘indirect DMFA’ is understandable in the context of this study (lines 351/362), the full acronym is ‘indirect direct membrane feeding’ is rather confusing.. I would suggest just using “indirect MFA” throughout the manuscript.

10) Line 245: Please elaborate clearly that these IC50 represents the IC50 of 3D7 gametocytes in the membrane feeding assay.

11) Line 245: Please elaborate what ‘duplicate’ means. Does it mean two feeders per isolate?

12) Table 1 was not available to reviewer (i.e. missing).

13) It would help reader to harmonize the order of compounds in Figures 5 and 6.

14) Line 154: Were dissection also done in plain distilled water? If not, please elaborate whether there is any impact of using water instead of PBS.

15) All analyses were on the infection rate. Because the numbers of oocysts were recorded (line 154), it would be good to analyze the data using the mean oocyst density (i.e. to determine the transmission-reducing activity rather than the transmission-blocking activity). This is probably a more linear/robust readout of % reduction of gametocyte infectivity.

16) Gametocyte viability assay: did it use early gametocytes or late gametocytes that were described in the parasite culture section?

17) Line 173: 3.5*104: superscript 4?

18) Discussion: A curious minds will want to know why incubation in whole blood for 24 hour led to loss of transmissibility. Would be nice to offer some speculation in the discussion. Was it possible that the parasites were simply eaten/destroyed by white cells?

19) Line 232: It may help to quickly state the effect of these compounds on 3D7 gam transmission (if data/references are available) to provide some mental calibration to understand where the data stand.

20) Line 290: “for during 24 hours at least”

Please soften the claim. There is not strict requirement for this. Although 24 hours is useful, someone might say 20 hours is sufficient.

21) Line 308: please provide reference for “In the absence of a pre-incubation with gametocytes, DHA, MM048 and Methylene Blue on did not affect infectivity of field gametocytes.”

22) Line 334, please reformat the reference.

23) Line 361 “this could explain our result”. Please elaborate further.

I agree that the effect on early gam could explain the lack of inhibition in Indirect-MFA using field isolates. But does this also mean that the blockage of 3D7 by DHA (as mentioned on line 255) was because the experiment was performed using early gametocytes? Readers would want you to help dispel the source of discrepancy between 3D7 and field isolates.

6. PLOS authors have the option to publish the peer review history of their article (what does this mean?). If published, this will include your full peer review and any attached files.

Reviewer #1: No

Reviewer #2: No

---

## [Author Response · Author response to Decision Letter 0]

6 Feb 2023

Changes in addition to the reviewer comments/editorial requests that are highlighted in the text:

Title: since we enrolled asymptomatic parasite carriers, we have removed the word patient from the title

Authors: the revision of the manuscript has resulted in a slight change in author order that all authors approve.

Abstract: we included some more specific results in the abstract. 

Introduction: we now included a short statement on recent findings that gametocytes can be artemisinin resistant. This gives an extra sense of urgency to our manuscript. We also updated the malaria morbidity and mortality figures with the latest WHO report. 

Methods: We added some general text to help the reader appreciate what work was done in Burkina Faso and what work was performed in the Netherlands.

Line 122. We added details on how we transported blood from the site of phlebotomy to the site of culturing/feeding (request reviewer 1)

Figure 1. We updated this figure to provide a comprehensive flow diagram of the different assays we developed (request reviewer 1).

In line 157, we clarified how we decided on the compound concentrations for field testing.

In lines 178-192, we provide additional detail on exactly how the assay was optimized, as requested by the reviewer 1.

In lines 195 and 202, we introduce the abbreviations TB-DMFA and SPORO-DMFA to help the reader understand in the later results how the two assays complement each other.

In line 234, we provide the requested detail on the statistical test used.

In the Results section, we have added small details (highlighted in yellow) to improve clarity without changing any of the results or analyses from the original submission. 

The legends to all figures have been improved for clarity.

Discussion 436-454, we have added a description of how our findings compare with findings from a group in Mali who published a manuscript on a similar methodology while our manuscript was under review at PLoS ONE. Their findings complement ours with some relevant differences in findings that we highlight in this revised discussion section.

 

Reply to editorial and reviewer comments

In our revision, we adhered to these guidelines

In our revision, we have included this statement.

In our revision, we have included this in the revised manuscript and in the cover letter. 

This work was supported by grants from the Dutch PDP fund, Medicines for Malaria Venture and Italian cooperation in Burkina Faso. The funders had no role in study design, data collection and analysis, decision to publish, or preparation of the manuscript.

We would like the funding statement to read as

‘This work was supported by grants from the Dutch PDP fund, Medicines for Malaria Venture and Italian cooperation in Burkina Faso. Teun Bousema and Katharine A. Collins are further supported by a European Research Council (ERC) Consolidator Grant to Teun Bousema (ERC-CoG 864180; QUANTUM). The funders had no role in study design, data collection and analysis, decision to publish, or preparation of the manuscript.

We have added this to the cover letter.

4. Please include your tables as part of your main manuscript and remove the individual files. Please note that supplementary tables (should remain/ be uploaded) as separate "supporting information" files

We have done as instructed

Reviewers' comments:

Reviewer's Responses to Questions

Comments to the Author

1. Is the manuscript technically sound, and do the data support the conclusions?

Reviewer #1: Partly

Reviewer #2: Yes

2. Has the statistical analysis been performed appropriately and rigorously? 

Reviewer #1: No

Reviewer #2: N/A

 We have improved clarity on the statistical tests we performed. This is now mentioned in the revised methods section and the relevant parts of the results/figure legends.

3. Have the authors made all data underlying the findings in their manuscript fully available?

 Reviewers' comments:

Reviewer #1: No

Reviewer #2: Yes

Data have been provided as supplemental file. In a single excel file we provide the exact data that underly the figures.

4. Is the manuscript presented in an intelligible fashion and written in standard English?

Reviewer #1: Yes

Reviewer #2: No

Based on this reviewer comment, we have carefully re-written the text and apologize for the typographical and grammatical errors in the initial submission.

5. Review Comments to the Author

Reviewer #1: The authors developed an indirect membrane feeding assay (indirect-MFA), where researchers can test transmission-blocking activity of antimalarial drugs with field gametocytes after 24-hour incubation, and compared IC50 of several drugs between the indirect-MFA and standard membrane feeding assay (SMFA) with laboratory-adapted NF54 parasites (a gold standard method at this moment). While their scientific approach is reasonable, the major results are missing in the current manuscript; i.e., the authors did not submit Table 1. Therefore, this reviewer cannot assess whether their statements are supported by the data. Furthermore, while the development of indirect-MFA is one of major points of this manuscript, the method section is not satisfactory. Thus, it is difficult for other researchers to use the new assay.

We have taken this comment very seriously and have provided additional detail on our methodology, including exact conditions of blood samples in the feeder. We also added a new Figure 1 that helps explain what assays were used. Our revised manuscript allows readers to replicate findings; the provision of the data underlying the figures further helps in data interpretation. 

Major comments;

1) Please submit Table 1, and show statistical results (e.g., Pearson correlation coefficients and p-values) to support the conclusion; good agreement in IC50 among Gametocyte viability assay, indirect-MFA and indirect-SMFA (in Line 266-268).

We have clarified our statistical tests. There was confusion in the original manuscript and no correlation coefficients were in fact presented. We apologize for this unclarity that arose from an early draft of a report. Our manuscript is now consistent throughout and includes details on the statistical tests and the cut-off for significance in the methods section and in the figure legends.

2) Method for indirect-MFA

Please describe the method in detail so that readers can use the assay. More specifically;

During the 24-h incubation

(a) medium; Was it pure RPMI (based on Line 202), or same as NF54 culture medium (including hypoxanthine, HEPES and sodium bicarbonate; Line 158-160), or something else?

(b) container; flask, tube or plate to keep the infected blood

(c) hematocrit; was the blood diluted at a certain ratio, or same ratio of RBC/serum in the original blood?

(d) IC50 was pre-determined to set 0.1x, 1x and 10x IC50 concentrations. But it is not clear how IC50 for each drug was estimated; IC50 in prevalence of infected mosquitoes or that for oocyst density? What assay (and which strain of parasites) was used to determine the value? Did you use a value from publication (if so, cite the paper), or did you use your own data?

We have now included full details on medium condition (a), the way in which infected blood was stored and how this method was validated (b), the source details and volumes of serum and RPMI (c). Moreover, we describe in detail how we decided on the IC50 estimates, these were based on prior SMFA experiments where oocyst density was used as read-out. We cited the relevant paper, that was from our own group (d)

For the feed,

(e) composition of feeding samples; If 100% of the liquid part was replaced by a malaria naïve serum after the 24-h incubation, then there was no room to mix the test drug. Was it a mixture of XX%v/v of human serum and XX%v/v of drug solution in RPMI? And what was the hematocrit of final feeding samples?

Final hematocrit was not determined but we provide an exact procedure, including volumes, in the revised manuscript.

Minor comments;

3) Can Indirect-MFA check gametocytocidal activity?

This is a relevant point and also made us realize that indirect MFA was not ideal as terminology. We have now clarified in the introduction, methods and Figure 1 what the different DMFA experiments detect. For instance, the legend to the new figure 1 now reads: ‘Natural gametocyte isolates were used for two distinct assays detecting the overall transmission-blocking activity of compounds by incubating them with gametocyte infected blood for 24 hours (2. TB DMFA) or sporontocidal activity by directly adding compounds to a gametocyte positive blood meal just prior to feeding (1. SPORO-DMFA). ‘

The indirect-MFA (now called TB-DMFA) detects the effect of compounds against gametocytes and the sporontocidal effect. We can differentiate between direct anti-sporogony effects and effects that are only apparent upon longer exposure (and are thus likely to be anti-gametocyte).

Since there was no evaluation for gametocytemia or exflagellation after the 24-h drug treatment, from the lower infection prevalence, we cannot tell whether the drug has sporontocidal activity, gametocytocidal activity, sterilizing gametocytes, or mixture of them. For example, if a drag kills gametocytes during the 24-h incubation (gametocytocidal), the final gametocytemia should be lower than that in the starting blood, but it was not evaluated. On the other hand, a drug may just reduce female fertilization activity (which is technically very difficult to measure) without killing them. In any case, it is impossible to tease out the mechanism of action from the described indirect-MFA method. Please fix all related text in Introduction, Results and Discussion sections.

We agree and have updated the text accordingly. All amended text related to this comment is highlighted. We have carefully described the extent to which our assay allows discriminating the effects on gametocytes from a sporontocidal effect, as well as limitations in differentiating between gametocyte killing and sterilizing effects.

4) “overnight” incubation?

People usually think “overnight” means ~10-12-h, not 24-h. To avoid the confusion, please do not use a word of “overnight” throughout the paper.

We agree and have updated the text accordingly. It was 24 hours; this is now also explained in the figure that depicts the workflow.

5) Gametocyte viability assays

Based on the method section (Line 161-163), two types of gametocytes (early and late gametocytes) were prepared. But I guess Gametocyte viability assays were performed only using late gametocytes. If so, where did the authors use the “early gametocyte”?

In addition, for clarity, please specify that the assay was done with NF54, such as, we performed full dose response gametocyte viability assays with “NF54” for selected compounds (Line 263-264).

We apologize for this mistake in the original agree and have updated the text accordingly. Stage III-V gametocytes were used in the gametocyte viability assay, as described in earlier publications.

6) What is “indirect SMFA”?

In Line 254, a term of “SMFA” is written, but another term “indirect SMFA” is seen in Line 267. Are they the same? If so, please use the same terminology to avoid a confusion. If different, please use different words, e.g., “direct SMFA” and “indirect SMFA”. In addition, I guess “indirect SMFA” means that an assay where NF54 parasites are pre-incubated with a test drug for 24 h before feed, but please clarify.

We agree with this and have updated the text. Based on this important comment, we have also chosen a different name for our assays: TB-SMFA to describe transmission blocking activity that may be caused by anti-gametocyte or by sporontocidal effects and SPORO-SMFA that is purely detecting anti-sporogony effects by adding the compound of interest directly to the blood meal without an incubation step. We believe that, with our figure 1 and repeated explanations of the differences between these assays, we have improved clarity and satisfied the reviewer.

7) Bars and error bars

For all figures, please explain what bars (e.g., average, median, geometric mean) and error bars (e.g., sd, sem, 95%CI) mean.

This has been clarified in the figure legends. All error bars were sd. Some figures are updated based on reviewer comments.

8) Fig 4

Same as Fig 3, “paired” data points (same donor’s blood were treated with DMSO or drug for 24-h) should be linked. Without the lines, readers cannot interpret the results. For example, the authors conclude that no inhibition by Atovaquone for one isolate (Line 312-313). But if the infection rate for the one isolate changed from ~70% (the highest point in DMSO) to ~30% (the highest point in 100 nM drug), then the conclusion should be different. In addition, a paired test should be used for the statistical analysis (please specify the name of statistical test). Once the authors do so, I’m afraid conclusion may change, e.g., there could be a significant reduction by Methylene blue (and also by DHA). Please revise the related text if needed.

We have clarified this and provided updated figure. The findings, in terms of statistical significance, did not change: the test was robust and already took into consideration the paired nature of the findings. We do, however, agree that connecting lines are highly valuable for data interpretation.

9) Typo

Line 173; 10 to the power of 4, not 104

This was updated

Line 173, 174 and 176; 30 microliter, not 30 mL

Updated

Line 247; to determine the “infection rates”, not “oocyst intensities” (no TRA data in this paper).

Updated

Line 307, take out “on”

Correct, this was updated

Line 320 and 334; the format of citation is wrong.

References and all typo’s have been rectified

Reviewer #2: This is paper describes a method to determine transmission reducing activity of test compounds against P. falciparum isolates. Overall the authors have done an admirable job. Testing field isolates are logistically challenging but very important.

I have comments below to improve the manuscript and make it more transparent to readers.

1) The writing could be polished by a native English user. In several places, the tense is in appropriate and the article is missing. Words such as mosquitoes/mosquitos or transmission blocking/transmission-blocking should be harmonized throughout the manuscript.

We appreciate this comment and have drastically improved the grammar and style of the revised manuscript. This prompted a complete revision of parts of the manuscript

2) There is no description of how the transmission blocking experiment was done on 3D7/NF54 gametocytes (i.e. the Indirect-SMFA). I am not sure if the data are original to the study, or referred to published work. Please provide the details of this experiment in the method (if original data), or in the discussion (if from previous studies).

This has been clarified. No SMFA was performed as part of this study but the DMFA experiments have been better explained and the names of the assays was updated to improve clarity. We now describe the assays as SPORO-DMFA to detect sporontocidal effects and TB-DMFA to detect general transmission-blocking effects.

I ask this because it is not clear whether indirect-SMFA is comparable to Indirect-MFA. Did the experiment use enriched gametocytes, and if so, early gametocytes or late gametocytes? Was the treatment also for 24 hours in the same culture medium? Without these details, it’s difficult to know whether the discrepancy in the results of 3D7/NF54 vs field isolates was due to biological (i.e. isolate-to-isolate) differences or the technical aspects of the assays.

3) Line 109: Were the participants treated for malaria after providing 9 ml blood?

Indeed, donors received treatment. This has been clarified.

4) Figures 1 & 2 are not clear. The arrow after the 24 hour incubation should point to the mosquito feeding cup.

We have clarified the figure and now have a single figure that explained the workflow.

5) Malaria naïve serum used for a) the 24 hr culture medium or b) plasma/RPMI replacement before MFA: was it from an AB blood group donor? Please clarify this in the method section.

The source of blood (European serum A) was clarified throughout the manuscript

6) Line 116: “In D0 DMFA” Should this be D1 DMFA?

The reviewer is correct that this was a mistake, the text has been updated. 

7) Under the method section: Please add statement that, for Indirect DMFA, naïve serum containing the test compound at the test concentration was used to replace the medium before membrane feeding.

This has been updated

8) Keywords: Anopheles coluzzii? I thought the experiments used An. gambiae throughout.

The reviewer is correct. This has been rectified.

9) Although the term ‘indirect DMFA’ is understandable in the context of this study (lines 351/362), the full acronym is ‘indirect direct membrane feeding’ is rather confusing.. I would suggest just using “indirect MFA” throughout the manuscript.

As indicated above, we actually renamed our assays to avoid Direct Direct Membrane Feeding Assay and Indirect Direct Membrane Feeding Assay. 

10) Line 245: Please elaborate clearly that these IC50 represents the IC50 of 3D7 gametocytes in the membrane feeding assay.

This has been clarified, with the appropriate references.

11) Line 245: Please elaborate what ‘duplicate’ means. Does it mean two feeders per isolate?

This has been clarified; it indeed indicates two feeders

12) Table 1 was not available to reviewer (i.e. missing).

This was a mistake and has been rectified in the revised manuscript.

13) It would help reader to harmonize the order of compounds in Figures 5 and 6.

We have updated the figures and legends.

14) Line 154: Were dissection also done in plain distilled water? If not, please elaborate whether there is any impact of using water instead of PBS.

Dissections were done in PBS; this has been clarified. 

15) All analyses were on the infection rate. Because the numbers of oocysts were recorded (line 154), it would be good to analyze the data using the mean oocyst density (i.e. to determine the transmission-reducing activity rather than the transmission-blocking activity). This is probably a more linear/robust readout of % reduction of gametocyte infectivity.

We have presented all data as % infected mosquitoes. This is the most informative outcome measure for field experiments. While it is true that oocyst density can also be used for mosquito feeding assays with high mosquito infection intensities, these high intensities were not achieved in our study. This has been clarified in the statistical analysis section.

16) Gametocyte viability assay: did it use early gametocytes or late gametocytes that were described in the parasite culture section?

17) Line 173: 3.5*104: superscript 4?

18) Discussion: A curious minds will want to know why incubation in whole blood for 24 hour led to loss of transmissibility. Would be nice to offer some speculation in the discussion. Was it possible that the parasites were simply eaten/destroyed by white cells?

19) Line 232: It may help to quickly state the effect of these compounds on 3D7 gam transmission (if data/references are available) to provide some mental calibration to understand where the data stand.

20) Line 290: “for during 24 hours at least”

Please soften the claim. There is not strict requirement for this. Although 24 hours is useful, someone might say 20 hours is sufficient.

21) Line 308: please provide reference for “In the absence of a pre-incubation with gametocytes, DHA, MM048 and Methylene Blue on did not affect infectivity of field gametocytes.”

22) Line 334, please reformat the reference.

23) Line 361 “this could explain our result”. Please elaborate further.

I agree that the effect on early gam could explain the lack of inhibition in Indirect-MFA using field isolates. But does this also mean that the blockage of 3D7 by DHA (as mentioned on line 255) was because the experiment was performed using early gametocytes? Readers would want you to help dispel the source of discrepancy between 3D7 and field isolates.

6. PLOS authors have the option to publish the peer review history of their article (what does this mean?). If published, this will include your full peer review and any attached files.

---

## [Decision Letter · Decision Letter 1]

28 Feb 2023

PONE-D-22-17171R1Assessment of the transmission blocking activity of antimalarial compounds by membrane feeding assays using natural Plasmodium falciparum gametocyte isolates from West-AfricaPLOS ONE

Dear Dr. SOULAMA,

Thank you for submitting your manuscript to PLOS ONE. After careful consideration, we feel that it has merit but does not fully meet PLOS ONE’s publication criteria as it currently stands. Therefore, we invite you to submit a revised version of the manuscript that addresses the points raised during the review process.

Thank you very much for the efforts to significantly improve this manuscript. However, the Reviewer 1 still have minor comments to further improve the manuscript. Please consider these comments and  prepare re-revised manuscript.

We look forward to receiving your revised manuscript.

Kind regards,

Takafumi Tsuboi

Academic Editor

PLOS ONE

Journal Requirements:

Reviewers' comments:

Reviewer's Responses to Questions

**Comments to the Author**

1. If the authors have adequately addressed your comments raised in a previous round of review and you feel that this manuscript is now acceptable for publication, you may indicate that here to bypass the “Comments to the Author” section, enter your conflict of interest statement in the “Confidential to Editor” section, and submit your "Accept" recommendation.

Reviewer #1: (No Response)

Reviewer #2: All comments have been addressed

2. Is the manuscript technically sound, and do the data support the conclusions?

Reviewer #1: Partly

Reviewer #2: Yes

3. Has the statistical analysis been performed appropriately and rigorously? 

Reviewer #1: No

Reviewer #2: Yes

4. Have the authors made all data underlying the findings in their manuscript fully available?

Reviewer #1: Yes

Reviewer #2: Yes

5. Is the manuscript presented in an intelligible fashion and written in standard English?

Reviewer #1: Yes

Reviewer #2: Yes

6. Review Comments to the Author

Reviewer #1: The authors replied to majority of my concerns appropriately, and clarity of the manuscript has been improved significantly. However, this reviewer thinks further minor modifications are required before publication.

1) Statements are not fully supported by the data

1-a) Fig 3 and interpretation of the results

First, data where zero prevalence in the DMSO control should be excluded from the statistical analysis (such as PYR with Donor 3 and 4), as we cannot tell whether there was not drug effect or not. Having said that, it is OK to include such data in the figure if the authors want to show how many assays had zero prevalence in the controls. Second, insignificant results by the Wilcoxon matched-pairs signed rank test are not interpreted appropriately (while the selection of the statistical test is reasonable). Based on the test, unless there are >5 pairs, the statistical results are always insignificant (i.e., even if each of all 5 paired data showed reduction from 100 to 0%, p=0.0625). The reason to see “significant” differences only in ATQ and P218 was because the two drugs were tested with 6 donors’ parasites. For MB, PYR, FQ and LUM, all donors’ parasites (4 out of 4) showed reductions in prevalence, so it is possible that the 4 drugs (at least some of them) could also show “significant” reductions if they were tested with 6 donors’ parasites. Line 303-307 and 422-425 should be rewritten, considering the limitation of the study design. In addition, “ns” in Fig 3 should be removed for drugs with <6 pairs. General readers think “ns” means no difference, instead of not enough statistical power.

1-b) Correlations among the three assays (Table 1, Line 358-360, and Line 475-476)

It is reasonable to conclude that there was a significant correlation between SMFA and TB-DMFA (Spearman rank correlation coefficient of 0.92 with p=0.0004, excluding DHA data). But the conclusion written in Line 475-476 is opposite. A fair conclusion from this study is that there is a strong correlation between the two assays, except for DHA. Second, there is no correlation at all between TB-DMFA (or SMFA) and gametocyte viability assay by a Spearman rank test (p=0.777). Therefore, Line 358-360 seems incorrect, unless the authors checked the correlation differently (in such a case, please specific how it was tested).

1-c) Line 339; not only DHA, Ferroquine showed > one-log difference in IC50 between the two assays

1-d) Line 357; non only atovaquone, SJ733 did not show full inhibition.

2) Table 1

Since comparing the three assays is the one of main point of this study, please add 95%CI of IC50 estimates, at least for TB-DMFA data. Showing the 95%CI helps readers to intuitively understand whether 38 (SMFA) and 167 (TB-DMFA) for MMV693183 are truly different, or within the error of estimates, for example.

In addition, please replace from “GCT-DMFA” to “TB-DMFA”

3) Exclusion of zero prevalence data from IC50 analysis (Fig 4)

Same as above point 1-a), such data should not be included for IC50 analysis. The authors might do so, but not written in the current text/figure/supplement

4) How to describe TB-DMFA

In Fig 1, please add “sporontocidal effect” for TB-DMFA (only “gametocyte effect” is written in the current figure). For clarity, at least in Line 326 and 345, please specify that the drug were added to the feeders as well. Readers, who do not read the method section carefully, could misunderstand that there was no drug in the blood samples which were fed to mosquitoes.

5) Replace (or take out) “overnight” incubation to “24 hour” incubation.

While the authors fixed the most of them, “overnight” are still seen in Line 39, 254 and 256

Reviewer #2: (No Response)

7. PLOS authors have the option to publish the peer review history of their article (what does this mean?). If published, this will include your full peer review and any attached files.

Reviewer #1: No

Reviewer #2: No

---

## [Author Response · Author response to Decision Letter 1]

5 Apr 2023

Journal Requirements:

The reference list is complete. The reference number 7 was not cite correctly and modify tob e cited correctly; it is cited now as “WWARN Gametocyte Study Group. Gametocyte carriage in uncomplicated Plasmodium falciparum malaria following treatment with artemisinin combination therapy: a systematic review and meta-analysis of individual patient data. BMC Med. 2016 May 24;14:79. doi: 10.1186/s12916-016-0621-7. PMID: 27221542; PMCID: PMC4879753.”

Reviewers' comments:

Reviewer's Responses to Questions 

Comments to the Author

1. If the authors have adequately addressed your comments raised in a previous round of review and you feel that this manuscript is now acceptable for publication, you may indicate that here to bypass the “Comments to the Author” section, enter your conflict of interest statement in the “Confidential to Editor” section, and submit your "Accept" recommendation.

Reviewer #1: (No Response)

Reviewer #2: All comments have been addressed

2. Is the manuscript technically sound, and do the data support the conclusions?

Reviewer #1: Partly

Reviewer #2: Yes

3. Has the statistical analysis been performed appropriately and rigorously? 

Reviewer #1: No

Reviewer #2: Yes

4. Have the authors made all data underlying the findings in their manuscript fully available?

Reviewer #1: Yes

Reviewer #2: Yes

5. Is the manuscript presented in an intelligible fashion and written in standard English?

Reviewer #1: Yes

Reviewer #2: Yes

6. Review Comments to the Author

Reviewer #1: The authors replied to majority of my concerns appropriately, and clarity of the manuscript has been improved significantly. However, this reviewer thinks further minor modifications are required before publication.

1) Statements are not fully supported by the data

1-a) Fig 3 and interpretation of the results

First, data where zero prevalence in the DMSO control should be excluded from the statistical analysis (such as PYR with Donor 3 and 4), as we cannot tell whether there was not drug effect or not. Having said that, it is OK to include such data in the figure if the authors want to show how many assays had zero prevalence in the controls. Second, insignificant results by the Wilcoxon matched-pairs signed rank test are not interpreted appropriately (while the selection of the statistical test is reasonable). Based on the test, unless there are >5 pairs, the statistical results are always insignificant (i.e., even if each of all 5 paired data showed reduction from 100 to 0%, p=0.0625). The reason to see “significant” differences only in ATQ and P218 was because the two drugs were tested with 6 donors’ parasites. For MB, PYR, FQ and LUM, all donors’ parasites (4 out of 4) showed reductions in prevalence, so it is possible that the 4 drugs (at least some of them) could also show “significant” reductions if they were tested with 6 donors’ parasites. Line 303-307 and 422-425 should be rewritten, considering the limitation of the study design. In addition, “ns” in Fig 3 should be removed for drugs with <6 pairs. General readers think “ns” means no difference, instead of not enough statistical power.

RESPONS: We have addressed this. In the text, we have clearly indicated that caution is required in interpreting no significance as evidence of no difference. This has been indicated in the legend to figure 3 (page 12, line 317-318: ‘Of note, the number of observations is small so ns lack of statistical significance should not be interpreted as evidence of no difference’), in the Results section (lines 300-303 ‘These experiments showed that Atovaquone almost completely inhibited infectivity at 100nM and also demonstrated high potency of P218. With our limited number of paired feeds, we observed no evidence for a statistically significant effect of DHA, Methylene blue, MMV390048, DDD107498, Pyronaridine, Ferroquine, or Lumefantrine on gametocyte infectivity at micromolar concentrations (Figure 3)‘ and the Discussion section (line 424-426 ‘Although the number of gametocyte donors was modest and subtle effects cannot be ruled out, our data indicate … ‘

Of note, pairs with zero percent prevalence in the DMSO condition have been excluded from the statistical analysis. This has been clarified in the Methods and Figure legends. Also, the ‘ns’ has been removed from figure 3. 

1-b) Correlations among the three assays (Table 1, Line 358-360, and Line 475-476)

It is reasonable to conclude that there was a significant correlation between SMFA and TB-DMFA (Spearman rank correlation coefficient of 0.92 with p=0.0004, excluding DHA data). But the conclusion written in Line 475-476 is opposite. A fair conclusion from this study is that there is a strong correlation between the two assays, except for DHA. Second, there is no correlation at all between TB-DMFA (or SMFA) and gametocyte viability assay by a Spearman rank test (p=0.777). Therefore, Line 358-360 seems incorrect, unless the authors checked the correlation differently (in such a case, please specific how it was tested).

RESPONS: We have rephrased this. ‘Comparison of IC50 values between gametocyte viability assay, TB-DMFA using field isolates and SMFA using NF54 laboratory strain shows broad agreement between the assays for Methylene Blue, MMV048, MMV693183, SJ733, Pyronaridine, DDD107498 and Lumefantrine while other compounds (notably DHA showed marked differences in activity against NF54 and field isolates.’ (line 359-363)

2) Table 1

Since comparing the three assays is the one of main point of this study, please add 95%CI of IC50 estimates, at least for TB-DMFA data. Showing the 95%CI helps readers to intuitively understand whether 38 (SMFA) and 167 (TB-DMFA) for MMV693183 are truly different, or within the error of 

estimates, for example.

RESPONS: we appreciate this comment but, since we have tested only three concentrations for each compound the logistic regression model used to determine IC50 values did not converge to the point where it was able to resolve confidence intervals. We have refrained from strong statements about differences and have indicated the limitation of this study in the revised discussion section (“Our current study tested a limited set of concentrations per compound. Although this allowed a broad comparison between transmission-blocking effects observed against field isolates versus laboratory studies, a more detailed analyses of more subtle changes in potency would benefit from more extensive dose-response analyses.”).

In addition, please replace from “GCT-DMFA” to “TB-DMFA”

RESPONS: This has been addressed.

3) Exclusion of zero prevalence data from IC50 analysis (Fig 4)

Same as above point 1-a), such data should not be included for IC50 analysis. The authors might do so, but not written in the current text/figure/supplement

RESPONS: Experiments with zero prevalence in the DMSO controls were excluded from the IC50 analyses and this has been clarified in the legend to table 1. New analyses of the Pyronaridine data (without the zero prevalence experiment) led to a small change in the estimated IC50 from 8,754 to 8,640 nM

4) How to describe TB-DMFA

In Fig 1, please add “sporontocidal effect” for TB-DMFA (only “gametocyte effect” is written in the current figure). For clarity, at least in Line 326 and 345, please specify that the drug were added to the feeders as well. Readers, who do not read the method section carefully, could misunderstand that there was no drug in the blood samples which were fed to mosquitoes.

COMMENT: we appreciate this important rectification. We have updated this figure and legend (‘Natural gametocyte isolates were used for two distinct assays detecting the overall transmission-blocking activity of compounds by incubating them with gametocyte infected blood for 24 hours (2. TB DMFA; concurrently detecting both gametocyte and sporontocidal effects) or…’). We also updated the information in Lines 328 ‘This experimental set-up detects the effect of compounds against gametocytes and/or its sporontocidal effects’.

5) Replace (or take out) “overnight” incubation to “24 hour” incubation.

While the authors fixed the most of them, “overnight” are still seen in Line 39, 254 and 256

COMMENT: this has been updated.

Reviewer #2: (No Response)

7. PLOS authors have the option to publish the peer review history of their article (what does this mean?). If published, this will include your full peer review and any attached files.

Do you want your identity to be public for this peer review? For information about this choice, including consent withdrawal, please see our Privacy Policy.

Reviewer #1: No

Reviewer #2: No

---

## [Editor Report · Decision Letter 2]

10 Apr 2023

Assessment of the transmission blocking activity of antimalarial compounds by membrane feeding assays using natural Plasmodium falciparum gametocyte isolates from West-Africa

PONE-D-22-17171R2

Dear Dr. SOULAMA,

We’re pleased to inform you that your manuscript has been judged scientifically suitable for publication and will be formally accepted for publication once it meets all outstanding technical requirements.

Kind regards,

Takafumi Tsuboi

Academic Editor

PLOS ONE
---

## [Editor Report · Acceptance letter]

12 Apr 2023

PONE-D-22-17171R2 

Assessment of the transmission blocking activity of antimalarial compounds by membrane feeding assays using natural *Plasmodium falciparum* gametocyte isolates from West-Africa 

Dear Dr. Soulama:

I'm pleased to inform you that your manuscript has been deemed suitable for publication in PLOS ONE. Congratulations! Your manuscript is now with our production department. 

Kind regards, 

on behalf of

Prof. Takafumi Tsuboi 

Academic Editor

PLOS ONE